# Ca^2+^- and Voltage-Activated K^+^ (BK) Channels in the Nervous System: One Gene, a Myriad of Physiological Functions

**DOI:** 10.3390/ijms24043407

**Published:** 2023-02-08

**Authors:** Carlos Ancatén-González, Ignacio Segura, Rosangelina Alvarado-Sánchez, Andrés E. Chávez, Ramon Latorre

**Affiliations:** 1Centro Interdisciplinario de Neurociencia de Valparaíso (CINV), Instituto de Neurociencias, Universidad de Valparaíso, Valparaíso 2340000, Chile; 2Programa de Doctorado en Ciencias, Mención Neurociencia, Universidad de Valparaíso, Valparaíso 2340000, Chile; 3Doctorado en Ciencias Mención Biofísica y Biología Computacional, Universidad de Valparaíso, Valparaíso 2340000, Chile

**Keywords:** K+ channels, BK channels, neurobiology, ion-channels, nervous system, neuronal excitability, synapsis

## Abstract

BK channels are large conductance potassium channels characterized by four pore-forming α subunits, often co-assembled with auxiliary β and γ subunits to regulate Ca^2+^ sensitivity, voltage dependence and gating properties. BK channels are abundantly expressed throughout the brain and in different compartments within a single neuron, including axons, synaptic terminals, dendritic arbors, and spines. Their activation produces a massive efflux of K^+^ ions that hyperpolarizes the cellular membrane. Together with their ability to detect changes in intracellular Ca^2+^ concentration, BK channels control neuronal excitability and synaptic communication through diverse mechanisms. Moreover, increasing evidence indicates that dysfunction of BK channel-mediated effects on neuronal excitability and synaptic function has been implicated in several neurological disorders, including epilepsy, fragile X syndrome, mental retardation, and autism, as well as in motor and cognitive behavior. Here, we discuss current evidence highlighting the physiological importance of this ubiquitous channel in regulating brain function and its role in the pathophysiology of different neurological disorders.

## 1. Introduction

Large-conductance calcium (Ca^2+^)- and voltage-activated potassium (K^+^) channels (BK, also known as Maxi-K or Slo1) are ubiquitously expressed in the body playing important roles in blood flow regulation, renal excretion, muscle contraction, circadian rhythm, and hearing [1,2,3]. BK channels are also abundantly expressed throughout the central nervous system in the cortex, basal ganglia, hippocampus, thalamus, and cerebellum, among other areas (Table 1) [4,5,6]. There, BK channels are strategically positioned on the plasma membrane and intracellular compartments in both neurons and glial cells [7,8,9,10,11,12,13]. Typically, activation of BK channels leads to a massive efflux of K^+^ ions that hyperpolarizes cellular membrane potential to regulate neuronal excitability and synaptic function, and, ultimately, motor and cognitive behavior [14,15,16,17]. Within a single neuron, BK channels are known to be expressed within different compartments, accomplishing different functions [18], including somatic regulation of action potentials [10,19] and regulation of neurotransmitter release at synaptic terminals [16,20,21,22,23,24]. BK channels are also present in dendritic compartments and neuronal spines [25], where they can limit the postsynaptic potential [25,26,27] or dendritic Ca^2+^ signals [25,28]. Although BK channels are also localized in nuclei, lysosomes, and mitochondria [11], their contribution to regulating neuronal function remains elusive.

While increasing evidence supports a critical role for BK channels in regulating neuronal excitability, their ability to act as a coincidence detector for changes in intracellular Ca^2+^ concentration suggests that BK channels are also well-positioned to control synaptic communication by exerting either negative [23,24] or positive [29] regulation of Ca^2+^ influx through voltage-activated Ca^2+^ channels (for review see Griguoli et al., 2016 [16]). However, the mechanisms by which BK regulate neuronal function and behavior in the brain remain far from being completely understood.

## 2. BK Biophysical and Structural Characteristics

Coded for by a single gene (*Slo1, KCNMA1*), BK channels have been proven, from the very beginning, to show a complex, albeit very attractive behavior. Their large conductance (≈250 pS in symmetrical 100 mM K^+^) near the limit of the diffusional process of K^+^ in solution is accompanied by an exquisite K^+^ selectivity, making them unique in the large family of different K^+^ channels [30,31,32]. Cloning of BK channels showed their similarity to voltage-dependent K^+^ channels in which each subunit contains six transmembrane domains, including a well-defined voltage sensor (S1–S4) and a pore domain (S5–S6) [33,34]. However, later studies showed that the channel-forming protein contained an extra segment (S0) placing the N-terminus on the extracellular side [35]. In addition, BK possess a large C-terminus, composed of two regulators of K^+^ conductance (RCK1 and RCK2), where the high-affinity Ca^2+^-binding sites reside [36,37,38,39].

Once single-channel, macroscopic, and gating currents records were available, one of the main challenges was to combine BK voltage dependence and channel activation by Ca^2+^ in one gating-kinetics scheme. From the start, it became clear that BK gating kinetics are intricate. One big step ahead was achieved by the group of Magleby [40], who identified that the several open and closed states present in the single-channel current records were correlated and inversely related. Shorter open intervals were followed by long closed intervals and vice versa. These observations made linear gating-kinetics models as those proposed, for example, for shaker K^+^ channels, unlikely. Instead, they suggested a progression of closed states of decreasing duration connected to open states of increasing lifetimes. More importantly, these findings hinted that the gating in BK channels is allosteric. Another important piece of information was that treatment of BK channels with *N*-bromoacetamide rendered BK voltage-dependent channels insensitive to internal Ca^2+^, a clear indication that voltage sensors and Ca^2+^-binding sites were in different regions of the channel-forming protein [41]. These early discoveries allowed us to define BK channels from the start as modular proteins with allosteric gating before knowing their well-defined structure.

The BK channel gating characteristics have been summarized in a two-tiered kinetic model [42,43]. Considering the tetrameric nature of BK channels in this allosteric gating-kinetics model, the four voltage sensor domains (VSD) undergo a transition between resting (R) and active states (A). The transition R–A is defined by a voltage-dependent equilibrium constant *J.* The VSDs allosterically interact with the pore opening through the allosteric factor *D* (Figure 1A). The model considers the existence of four Ca^2+^-binding sites, each of which includes a Ca^2+^-binding mediated transition between an empty site (U) and a Ca^2+^-bound site (BCa^2+^) with an equilibrium constant K = [Ca^2+^]/K_D_. In this case, the allosteric interaction with the pore opening is determined by the allosteric factor *C.* On the other hand, the strength of allosteric interaction between voltage sensors and Ca^2+^ binding is determined by the allosteric factor *E*. It is important to note here that the closed–open (C–O) reaction defined by the equilibrium constant *L* can proceed in the absence of voltage-sensor activation and of internal Ca^2+^, albeit with a very low probability of opening (Figure 1B). Figure 2B shows that Ca^2+^ alone can increase the open probability (Po) by about four orders of magnitude. Since Po increases with depolarizing voltages and internal Ca^2+^ concentrations, and BK has large conductance, BK channels are the perfect damping machine to brake excitatory processes mediated by internal Ca^2+^.

There is a strong coupling between the Ca^2+^ binding sites and the voltage sensors. At saturating Ca^2+^ concentrations, the equilibrium constant *J* that defines the resting–active equilibrium of the voltage sensor (Figure 1A) increases 26-fold [44,45]. This result implies that the free energy necessary to activate the BK channel voltage sensor decreases by about 8 kJ/mol at saturating internal Ca^2+^ concentrations (100 µM).

The modular nature of BK channels was confirmed when the first BK structures were determined using cryo-microscopy [38,46]. As voltage-dependent K^+^ (Kv) channels, BK channels are tetramers composed of four identical α subunits (Figure 2A), but unlike Kv channels, BK channels have a non-swapped configuration. The VSD of one subunit (S1–S4) contacts the pore domain (S5–Pore domain–S6) of the same subunit (Figure 2 A, C). Although the work of Ma et al. [47] suggested the presence of a decentralized VSD in which the gating charges were spread across S2, S3, and S4, more recent work gave strong evidence that the gating charges are two arginines contained in S4 (R210 and R213) located in a septum devoid of water [48]. The movements of the charges during BK-VSD activation are small, with only a modest (≈2 Å) displacement of S4. It is unclear at present how the electrical energy is transformed into opening of the pore. However, the gating ring probably mediates the coupling between VSDs and the pore [38,45,46].

The two RCK domains form a gating ring (Figure 1B) containing the two high-affinity Ca^2+^-binding domains that can be distinguished structurally by their selectivity to divalent ions. The divalent cation-binding site in RCK1 preferentially binds Ca^2+^ and Cd^2+^, whereas in RCK2, the preferred divalent cations are Ca^2+^ and Ba^2+^ [38,49,50]. The binding of Ca^2+^ promotes an expansion of the gating ring, which, by pulling the linker that connects the gating ring to S6, causes this transmembrane segment to adopt a configuration that leads to channel opening [46]. It is unclear at present where the gate that defines the closed–open transition is located (Figure 1A). The BK structures show that there is only a modest decrease in the radius of the internal BK vestibule when the channel closes, which varies between 1 and 4 Å [38]. The diameter of the narrowest part of the vestibule is 12 Å. If there is no well-defined structure hindering ions’ passage through the BK channel, the following question arises: Where is the gate? Blockade mediated by large quaternary ammonium ions gave the first hint suggesting that, similarly to other channels [51,52], the BK gate resides in a selectivity filter [53,54,55]. However, simulations of molecular dynamics using BK structures in the absence and the presence of Ca^2+^ indicated that the actual decrease in diameter of the internal cavity of the BK channel during closed state is larger than the one shown by the BK structure obtained in the absence of Ca^2+^ [56]. The final diameter of the cavity was 6–8 Å, and its walls became more hydrophobic, creating a region devoid of water. Due to its low dielectric constant, it produces an energy barrier unsurmountable to ions.

## 3. BK Auxiliary Subunits

The four auxiliary β subunits (β1–β4) in BK are characterized by a common structure consisting of two transmembrane segments connected by a large extracellular loop (~148 aa residues) and cytoplasmic COOH and NH2 termini. Although to a different degree, all β subunits modify BK channel gating and pharmacology (reviewed in [10,57,58]), β1, β2, and β4 increase Ca^2+^ sensitivity by stabilizing the voltage sensor’s active conformation and slowing down the activation and deactivation kinetics. Mediated by its NH_2_ terminus, β2 also induces a fast and complete inactivation. Four isoforms are generated by splicing the β3 gene (*kcnmb3*; β3a–d). β3 subunits do not modify BK channel Ca^2+^ sensitivity, but three of them (β3a–c) produce a fast and incomplete inactivation. Importantly, β subunits have different pharmacological profiles, as β2 is iberiotoxin (IBTX)-sensitive, whereas β4 is IBTX-resistant. However, BK channels consisting of the complexes α/β2 or α/β4 can be blocked by paxilline [10,57,59,60]. β4 slows down the activation and deactivation kinetics at all Ca^2+^ concentrations. In symmetrical K^+^-concentration conditions, β4 promotes a leftward shift of the BK conductance–voltage curve only at Ca^2+^ concentrations >10 µM compared to BK channels formed by the α subunit alone [61,62]. However, here it is essential to highlight the fact that under physiological conditions—low external K^+^—BK/β4 channels shift the conductance–voltage curves leftward at all internal Ca^2+^ concentrations compared to BK/α channels [63,64].

When characterizing the different types of Ca^2+^-activated channels in the brain plasma membrane vesicles, Reinhart et al. [65] detected two types of high-conductance BK channels. One type showed fast gating and was blocked by charybdotoxin (ChTX) (type I), and the other was ChTX-resistant and exhibited slow gating (type II). Cloning of the different β subunits (β1–β4) showed that the toxin resistance of some BK channels in the brain was most probably due to the formation of complexes between the pore-forming α and the β4 subunit [60,66,67]. Swapping the long external loop connecting TM1 and TM2 between β4 and β1 and neutralization of some of its basic residues showed that this region of the β4 subunit hinders the binding of iberiotoxin (IBTX) by blocking access of the toxin to the BK external vestibule [60,68]. The cryo-structures of the α/β4 BK channel show a 1:1 stoichiometry between α and β4 subunits in which the four external loops of the β4 give origin to a crown-like structure [69]. The crown impedes both access to and toxin release from the binding site, explaining the extremely slow ON and OFF rates of IBTX-inhibition kinetics [70].

β4 is the most abundant auxiliary subunit expressed in the nervous system [66,71]. β4 is found in the thalamus, brainstem, posterior pituitary terminals, pyramidal neurons of the cortex, CA3 pyramidal neurons in the hippocampus, hippocampal dentate granule cells, olfactory bulb, and Purkinje cerebellar neurons [64,72,73,74,75] (Table 1). β2 is also extensively expressed in the brain, albeit at lower levels than β4 [66,71]. β1 on the other hand, is almost absent in the brain, but it has been detected in the hypothalamus, paraventricular neurons, and cerebellar Purkinje cells [76,77,78]. Interestingly, these hypothalamic neurons reportedly express β1 subunits on the soma and dendrites, while the axon terminals mainly contain the β4 subunit [79]. Such specific distribution of β subunits within the same neurons may have important functional significance in understanding how BK channels differentially contribute to the regulation of somatic and nerve terminal compartments.

The physiological role of the α/β4 BK channels in the brain was first elucidated by Brenner et al. [72], who showed that this type of BK channel (type II) is present in the dentate gyrus granular cells regulating their excitability. Knockout of the β4 converted type II into type I (fast) BK channels [72]. Type I channels reduce, and type II BK channels increase the duration of action potentials [63,72]. Interestingly, in pathological conditions, such as a neuropathic pain model, β4 seems to be upregulated in anterior cingulate cortex pyramidal neurons [73], suggesting that the expression of BK channel subunits could be modulated in an activity-dependent and cellular-specific fashion. Moreover, evidence suggests that the fragile mental retardation protein (FMRP), by interacting with the β4 subunit, shortens the action potential, thus regulating neurotransmitter release in CA3 pyramidal neurons [15]. Single-channel recording in CA3 pyramidal neurons showed that the α/β4 BK channel open probability is reduced by about 50% in neurons from *fmpr* knockout mice increasing the inter-burst interval [80]. However, a recent study showed that FMPR modulates both α/β4 BK and α BK channels [81]. The degree of importance in modulating synaptic activity via interaction of FMPR with α/β4 BK or α BK channels is currently unclear.

Inactivating BK currents have been detected in mammalian auditory hair cells, rat lateral amygdala neurons, and hippocampal CA1 neurons [82,83,84]. In CA1 and dorsal root ganglion neurons, inactivation is removed when the inner membrane surface is treated with trypsin, suggesting that BK channels are formed by the α/β2 complex [84,85]. Consistent with this idea, in rat dorsal root ganglion neurons, α/β2 BK channels are blocked by IBTX, resulting in increasing firing frequency, broadening of the action potential, and reducing afterhyperpolarization [85]. The spontaneous firing rate in the suprachiasmatic nucleus (SCN) exhibits daily oscillations that determine the normal circadian timing of behavior in mammals. The intrinsic circadian clock in the SCN controls the daily expression of BK. Consistently, BK currents are lower during the day and high at night [86,87,88], thus determining the daily SCN rate of firing. Moreover, inactivating BK currents are predominant during the day and are reduced at night with an increase in steady-state BK currents [86]. Importantly, in the SCN, BK inactivation is mediated by the β2 subunit, and β2 knockout mice show a lack of diurnal variation in BK currents and SCN firing rate [86].

## 4. BK Channel Coupling with Ca^2+^-Permeable Channels

**Ca_V_ and BK Nanodomains:** Critical for the feedback control of Ca^2+^ influx and cell excitability in the nervous system, BK channels co-localize with voltage-dependent Ca^2+^ (Ca_V_) channels and N-methyl-D-aspartate receptors (NMDAR; reviewed in [89,90]). Due to the low affinity of BK for Ca^2+^ (Kd ≥ 3 µM; [91]), co-localization with Ca^2+^-permeable channels provides BK channels with an effective local Ca^2+^ concentration for their activation and function [92,93]. Indeed, BK channels have been reported to form nanodomains with L-type, P/Q, N-type, R-type, or T-type Ca_V_ [94,95,96,97,98].

Ca^2+^ chelator EGTA and fast chelator BAPTA differ in their Ca^2+^-binding rates but have similar Ca^2+^ affinities. They have been a powerful tool for determining the diffusion distance for Ca^2+^ from calcium channels to BK channels. BK channel activity was abolished when using BAPTA as the internal Ca^2+^ buffer, but not with EGTA, revealing the close spatial proximity (~10 to 20 nm) in the nanodomains between BK channels and voltage-sensitive calcium (Ca_V_) channels [96,99,100]. More recently, a mathematical model revealed the coexistence of L-type and BK-type Ca_V_ channels and suggested that they are grouped in nanodomains spatially separated 30 nm from each other [101].

Activation of BK channels mediated by the influx of Ca^2+^ through Ca_V_ channels decreases the action potential duration, promotes fast hyperpolarization potentials, decreases the time of Ca^2+^ influx, and limits the release of neurotransmitters [99,102,103]. The activation of Ca_V_ channels modifies the voltage dependency, current amplitude, and kinetics of BK channel activation, causing functional variability in BK dependent on the expression pattern of Ca_V_ channels [96,104]. The pharmacological blockade of BK channels using IBTX, ChTX, or paxilline has also demonstrated the functional coupling between BK and Ca_V_ channels. The application of these toxins increases the amplitude of the action potential and Ca^2+^ entry through Ca_V_ channels as well as the release of neurotransmitters [99,102,105,106,107,108]. On the other hand, the specific blockade of a subset of Ca_V_ channels suppresses the hyperpolarization that is provided by the flow of K^+^ ions from active BK channels [96,97,102,105,106,109,110]. One example that illustrates the importance of Ca_V_ and BK localization is the functional and spatial localization of Ca_V_1.3 and BK channels in rat hippocampal and sympathetic neurons [104]. Ca_V_1.3 activates at low voltages, and when co-expressed with BK, the latter started activating at about −50 mV. In hippocampal and sympathetic neurons, Ca_V_1.3 and BK are localized as clusters of Ca_V_1.3 encircling clusters of BK channels, forming a multi-channel complex. Since BK channels activate at voltages near the action potential threshold in these complexes, they enable the modulation of neuronal excitability.

**BK and NMDAR Complexes:** BK channels and NMDARs are widely expressed in the brain, and subcellular distribution variations are caused by the appearance of distinct BK channel auxiliary subunits and NMDAR subunit composition. So far, the BK–GluN1 complex has been functionally described on granule cells of the olfactory bulb [111], on mature dentate granule cells in the hippocampus [27], and in a subset of layer 5 pyramidal neurons of the barrel cortex and visual cortex [25]. Moreover, pulling-down assays demonstrated that this complex is also present in the cerebellum, cortex, thalamus, and striatum [27].

Functional coupling between NMDARs and BK channels was first demonstrated in outside-out patches from the cell bodies of granule cells, where the glutamate-induced outward currents were abolished after applying TEA, paxilline, or IBTX [111]. Distinctively, in the BK–GluN1 complex, the BK S0–S1 loop interacts with the C-terminal domain of the NMDARs. NMDAR activation by glutamate produced robust BK outward currents in whole-cell voltage-clamp recordings on mature dentate gyrus granule cells [27]. The application of NMDAR antagonist (2R)-amino-5-phosphonovaleric acid (AP5) and the pore-blocker MK-801 inhibited the outward currents, confirming the tight association of the BK–GluN1 complex. BK channel outward currents were also elicited by NMDAR activation in neocortex pyramidal neurons [112], those BK currents were also eliminated by AP5, confirming that NMDAR activation is needed to trigger the opening of BK channels. Likewise, ZnCl_2_ and ifenprodil (IFEN) were used to inhibit GluN2A or GluN2B-containing NMDARs, respectively, to assess the selective coupling between BK channels and different NMDAR subunit compositions. Nevertheless, these compounds only partially suppressed the BK outward currents, suggesting that GluN2A or GluN2B might also be a part of the functional coupling between BK channels and NMDARs in the brain.

NMDAR–BK coupling efficiency with the different NMDAR subunits was also tested in HEK cells expressing BK and GluN1/GluN2A or GluN1/GluN2B using a proximity ligation assay. In agreement with the pharmacological data, BK channels showed no preference for different NMDAR subunits [112]. To further evaluate the degree of coupling between BK channel activation and NMDAR, the Ca^2+^ chelators EGTA and BAPTA were used to determine the relative distance between BK channels and NMDARs in granule cell bodies [111]. The BK channel outward currents induced by glutamate were insensitive to the applied intracellular EGTA. In contrast, BAPTA abolished the BK currents induced by the activation of NMDARs. BK–NMDAR coupling has also been tested in hippocampal and pyramidal neurons using Ca^2+^ chelators [27,112]. In both cases, the results show that BK is robustly activated by the Ca^2+^ influx mediated by NMDAR, suggesting a short coupling distance.

**Other Sources of Ca^2+^ for BK Channel Activation:** Co-localization of BK channels with Ca_V_ channels may form a dynamic and reversible channel complex between the plasma membrane and the endoplasmic reticulum. In neurons of the cerebellum, hippocampus, and suprachiasmatic nucleus, the Ca^2+^ influx mediated by Ca_V_ channels activates the ryanodine receptors (RyR) present on a nanodomain of the endoplasmic reticulum. RyR release sparks caused by Ca^2+^ immediately below the plasmatic membrane, result in generating rapid activation of the BK channels, forming a complex commonly named a Ca_V_–RyR–BK triad [113,114,115]. This triad allows control over action potential firing patterns on a millisecond time scale [113,114]. Moreover, interaction of BK channels with transient receptor potential vanilloid receptor 1 (TRPV1) channels has also been demonstrated in rat dorsal root ganglion neurons and in heterologous systems where both channels were co-expressed. Using electrophysiological recordings and co-immunoprecipitation assays showed that BK channels are activated by Ca^2+^ influx through TRPV1 channels, with which they associate in complexes in a largely EGTA-insensitive manner [116].

## 5. BK Post-Translational Modifications

In addition to the diversity of subunits controlling BK function, extensive post-translational modifications significantly modify channel function [117,118,119,120]. These post-translational modifications impact the contribution of BK channels to the control of neuronal function. One of the most studied modifications is the addition of phosphate groups to functionally essential residues (Ser/Thr/Tyr) present within the BK α subunit [9], where at least 30 Ser/Thr phosphorylation sites have been identified. Changes in phosphorylation status differentially modulate the voltage and Ca^2+^-dependent activation [118]. This effect seems to be developmentally regulated [121] as phosphorylation by protein kinase A (PKA) and protein kinase C (PKC) are inversely predominant during fetal and adult stages [117]. While phosphorylation can suppress BK currents induced by an action potential [122], inhibition of phosphatases or mutation in Ser/Thr phosphorylation sites hinders the slow-down of the BK gating kinetics induced by the β4 subunit [123]. These post-translational modifications of BK channels imply that different functions might arise depending on their phosphorylation status. For instance, in medial vestibular nucleus neurons, BK channels contribute to afterhyperpolarization in a CaMKII-dependent manner [124]. In contrast, PKC decreases BK channel activity [125]. In cerebellar Purkinje neurons, PKA can activate non-inactivating BK channels with low activity but inhibits channels with higher activity [126]. PKCγ on the other hand, negatively modulates BK currents resulting in attenuation of the electrical signal and significant alterations of the complex spike waveform [127]. In presynaptic terminals, activation of BK channels and their subsequent phosphorylation by CaMKII modulates neurotransmitter release [119,128], synaptic depression underlying habituation [129], and input-specific spike-timing-dependent long-term depression at thalamostriatal, but not at corticostriatal synapses [28]. Similar to kinases, phosphatases can also regulate the function of BK channels mainly through an inhibitory control of the channel [106,130,131,132]. Accordingly, type I BK channels lacking β4 can increase neuronal excitability in dentate granule cells in response to reduced phosphatase activity [133].

## 6. BK Channels Differentially Contribute to Control of Brain Function

The BK α subunit is highly expressed during late embryonic and early postnatal development [14]. As development progresses, BK currents undergo an abrupt increase during the first weeks after birth in neocortical pyramidal neurons, substantia nigra dopamine neurons, and cochlear inner hair cells [134,135,136]. Similarly, BK channel expression in hippocampal neurons is likely to be developmentally regulated, differentially contributing to the repolarization of single action potentials during the first postnatal week [137], suggesting that BK channels contribute to shaping the properties of mature neuronal firing. One of the clearest roles of BK channels in regulating neural activity comes from electrophysiological studies of neuronal firing. Several reports showed that BK channels control action potential shape by contributing to repolarization and fast afterhyperpolarization (fAHP) currents in multiple neuronal types. These include cerebellar Purkinje cells [102,138,139,140], hippocampal pyramidal cells [141,142,143], cortical pyramidal neurons [144,145], and striatal medium spiny neurons [146,147] (Table 1). As discussed above, the strategic co-localization of BK with Cav channels allows BK channels to control Ca^2+^ influx during action potential repolarization and fAHP, even in spontaneously firing neurons [34], suggesting that BK channels act as a feedback regulator to control neuronal excitability. While the BK–Cav channel coupling indicates that BK channels participate during the repolarization phase of the action potential, their coupling with the RyR contributes to the free Ca^2+^ necessary for BK channel activation during the fAHP [148]. However, recent evidence in CA1 pyramidal neurons indicates that the BK channel–Cav2.3 complex regulates somatic excitability by controlling fAHP rather than repolarization. Cav2.3 KO mice showed increased excitability with a decreased fAHP [103]. Such differences may arise from the non-homogenous distribution of BK channels in clustered or scattered pools. While clustered pools may be involved in slower events triggered by Ca^2+^ released by intracellular domains, scattered pools could be involved in transient events such as spike repolarization [139]. Adding to the complexity of BK channel function regulating neuronal firing, evidence indicates that BK channels could also increase neuronal firing [142]. By accelerating membrane repolarization and limiting sodium channel inactivation, BK channels can promote repetitive spiking [149]. An enhanced fAHP can also increase neuronal firing by promoting sodium-channel recovery from inactivation [150].

In some neurons, such as Purkinje [177] and vestibular neurons [178], inhibiting BK channels speeds up action potential firing accompanied by a reduction in afterhyperpolarization. Paradoxically, in substantia nigra dopaminergic neurons, inhibiting BK channels increased spike width. This effect was accompanied by an increase in afterhyperpolarization current, likely involving an enhancement of the slowly deactivating K_v2_ current [179]. Similarly, in vestibular nucleus neurons, the fastest firing neurons express Kv3 channels, predominantly to repolarize the neuronal membrane during an AP train. However, in slower-firing neurons, BK channels are responsible for repolarization [180]. This evidence supports the idea that the functional role of BK channels cannot be defined in isolation but critically depends on the context of the other conductances in the cell, as well as their precise localization and composition.

As for principal cells, BK channels also have been reported to control excitability of GABAergic interneurons, but in a lesser extension. For example, in the cortex, action potential repolarization is governed by BK channels in somatostatin (SST)- but not Parvalbumin (PV)-positive interneurons [166]. Interestingly, in the striatum, SST enhances BK currents [146]. If SST released from SST-positive interneurons could act in an autocrine manner to regulate BK channels and control their excitability remains unknown. It is noteworthy, however, that in isolated retinal bipolar cells, SST inhibits BK channels [181]. BK channels also regulate action potential repolarization in striatal cholinergic interneurons [21], regulate firing properties of GABAergic vestibular nucleus neurons [182], and contribute to fAHP in cholinergic neurons of the pontine laterodorsal tegmentum [183]. In the retina, BK channels suppress activation of Ca^2+^ channels to regulate GABAergic A17 amacrine cell signaling [22]. On the other hand, in fast-spiking stellate cerebellar interneurons, blocking BK channels increases action potential width and Ca^2+^ influx [184], further supporting a role for BK in regulating GABAergic interneurons function.

**BK channels regulate dendritic excitability:** In addition to controlling somatic excitability, increasing evidence indicates that BK channels are also present in dendritic compartments [5], where they exert an additional level of control by modulating dendritic excitability. For example, in the somatosensory cortex, BK channels are present in apical dendrites [154], exerting a suppressive effect on the Ca^2+^ channel to regulate Ca^2+^ spikes and dendritic excitability [18,144,185]. This control is activity-dependent, as a single action potential or a train of low-frequency action potentials could not activate BK channels [156]. In contrast, a high-frequency train of action potentials can activate BK channels [185]. BK channels also restrict dendritic Ca^2+^ spikes in CA1 pyramidal cells, thereby generating an inhibitory effect over synaptic potentiation [186] although dendritic BK channels are not activated by backpropagating action potentials [158,187]. Inhibition of BK channels was also shown to improve dendritic Ca^2+^ spikes in cerebellar Purkinje neurons, thereby facilitating short-term synaptic plasticity mediated by endocannabinoids at distal spines [171] or signal propagation from dendrites to the soma [127]. BK channels also control the magnitude of excitatory potentials in cerebellar Purkinje dendrites [172]. BK-mediated effects for dendritic spike generation could depend on the spine size where they are expressed. For example, in basal dendrites of pyramidal neurons of layer V in the visual cortex, BK channels are activated only in small-head spines, where they suppress excitatory postsynaptic potentials [25]. This effect requires a BK–NMDA receptor complex rather than coupled BK–Cav channels, as Ca^2+^ influx through NMDAR is required to activate BK channels [25]. A similar coupling between NMDAR and BK channels was also observed in basal dendrites of layer V pyramidal neurons in the barrel cortex, where they avoid the induction of spike-timing-dependent plasticity [188]. However, the BK–NMDAR complex is not observed in all basal dendrites from cortical neurons, where coupling between SK channels and NMDARs to regulate dendritic excitability has also been described [185]. Interestingly, BK-mediated regulation of dendritic excitability in layer 5 neocortical neurons is impaired in mice lacking FMRP [151], thereby suggesting that BK channels might play a role in the pathophysiology of mental deficiency.

**BK channels regulate synaptic transmission:** BK channels also appear to be essential regulators of neurotransmitter release when expressed at synaptic terminals within different brain circuits [14,16]. Their localization in the active zone appears to be regulated by different synaptic proteins, including RIM-binding proteins [189], synapsin [190], α-catulin [191], and dystrobrevin [191,192]. In the brain, the function of presynaptic BK channels in regulating neurotransmitter release may vary from synapse to synapse [23,24]. For example, BK channels regulate excitatory glutamate-release probability, an effect associated with broadening a presynaptic action potential at CA3–CA3 connections [24]. In contrast, at Schaffer collaterals to CA1 synapses, BK channels contribute to transmitter release only after action potential broadening with 4-aminopyridine, and thus, it has been suggested that BK channels act as ‘emergency brakes’ to exert a protective effect against synaptic hyperactivity [23]. A similar effect was observed at glutamatergic mossy fiber boutons in rat hippocampal slices, where the contribution of BK channels to presynaptic action potential repolarization was not observed under basal conditions due to the presence of a subtype of fast-activating Kv3 channels [164]. Moreover, evidence suggests that postsynaptic BK channels in mature dentate granule cells [27] and thalamocortical circuits [112], could also regulate synaptic transmission via NMDAR-mediated channel activation. Interestingly, NMDARs could be localized at some presynaptic terminals, where their activation regulates transmitter release and plasticity [193,194,195]. Whether a similar BK–NMDA complex is present at presynaptic terminals, thereby regulating transmitter release, remains unknown.

BK-mediated control of synaptic transmission also has been described in rod photoreceptors in the salamander retina [29], and in the brainstem, where they regulate paired-pulse ratio and depression induced by trains of high-frequency stimulation [129]. Although a preferential control exerted by BK channels on excitatory versus inhibitory synapses has been suggested [196,197], it has become clear that BK channels could also control GABA release in different neuronal circuits. For example, in the retina [22] BK channels contribute to GABA release from A17 amacrine cells. In the cortex BK control GABA release from fast-spiking interneurons to II/III layer pyramidal neurons [21] and at GABAergic synapses in the central amygdala [12]. Likewise, BK channels regulate inhibitory glycine release in the spinal cord [198] and acetylcholine release from efferent terminals from the medial olivocochlear system to inner hair cells [199]. These results suggest that by regulating excitatory and inhibitory transmitter release, BK channels might contribute to the excitatory–inhibitory balance necessary for normal brain function. Consistent with this idea, it has been observed that cereblon, a protein related to intellectual disability, downregulates BK channel activity at hippocampal CA3 terminals affecting paired-pulse ratio and short-term plasticity, but not long-term plasticity [200].

**Glial BK channels:** In addition to being present in neurons, BK channels are also localized in astrocytes [201], where their activation reportedly modulates blood pressure [202,203] and generates vasodilation [204]. Interestingly, astrocytic BK channels mainly express the β4 subunit during neonatal development [12,205], thereby regulating voltage gradients and K^+^ homeostasis [12], but their contribution to regulating synaptic function remains unclear. In glioblastoma, the most invasive and aggressive primary brain tumor, astrocytic BK channels are likely upregulated to induce water loss and reduce cellular volume [206]. More recently, an interaction between BK channels and the GABA transporter type 3 (GAT3) has also been reported, suggesting that BK channels could regulate GABA homeostasis and signaling by regulating astrocyte function [207]. However, the role of astrocyte BK channels in regulating glutamatergic and GABAergic synaptic function remains unknown.

BK channels are also localized in the spinal cord [208] and cortical microglia [209]. In the spinal cord, BK can induce microglial hyperactivity related to neuropathic pain [210] and potentiate spinal synapses in lamina I spinal neurons [8]. In the brain, evidence suggests that microglial BK channels might participate in BDNF release and induce microglial phagocytosis [209]. BK channels have also been identified in microglial cells of the hippocampus [211,212], which, unlike neuronal BK, might not be developmentally regulated [213]. Whether microglial BK channels may contribute to regulate neuronal excitability and synaptic transmission within selective brain circuits remains unknown.

Lastly, BK channels also seem to be present in oligodendrocyte precursors [214,215] and to a lesser extent in mature oligodendrocytes [214]. In multiple sclerosis lesions, the β4-subunit coding gene KCNMB4, is upregulated in oligodendrocytes [216], but its role is largely unknown.

**Could BK channels be activated by endogenous ligands to regulate neuronal function?** In addition to being activated by voltage and Ca^2+^, increasing evidence suggests that different molecules can directly bind to BK channels to regulate their function. For instance, 17β-estradiol activates BK channels via a process that requires the presence of the β1 subunit [217]. Moreover, β4 confers BK channel sensitivity to corticosterone [218]. While evidence suggests that estradiol could regulate hippocampal excitability [219] and regulate glutamatergic synaptic transmission and plasticity in the hippocampus [220] and primary auditory cortex [221], whether these effects are directly mediated by targeting BK channels remains unclear.

Different lipid ligands interact directly with BK channels. For example, the omega-3 fatty acid DHA activates β1 and β4, with minimal effect on β2-containing BK channels [222]. Similarly, arachidonic acid (AA) increases BK channel currents in the presence of β2 subunits [223] and AA metabolites produced by a COX-mediated pathway are responsible for BK channel activation in microglia [8]. In vascular tissues, lipoxygenase (LOX)-metabolites have also been shown to activate BK [224]. In particular, BK channels from coronary artery smooth muscle, which contains β1 and β4 subunits, are activated by AA-lipoxygenase metabolites [225].

The endocannabinoid anandamide and its stable analog, methanandamide, also activate BK currents [226,227]. Moreover, a direct interaction between cannabinoid signaling and the BK channel appears to be involved in the effect of cannabinoids suppressing peripherical firing in injured fibers [228]. However, it is unknown whether endocannabinoids can regulate brain function by targeting BK channels.

## 7. BK Channel Dysfunction in Neurological Disorders

Given the critical role of BK channels in regulating neuronal excitability and synaptic function, it is not surprising that dysfunction of BK channels has been implicated in several neurological disorders. These include epilepsy, fragile X syndrome, mental retardation, autism, movement disorders, and chronic pain [229]. The use of BK channels in animal models of BK loss-of-function (LOF) and gain-of-function (GOF) mutants has increased our current understanding of the role of BK channels in the etiology of these disturbances. For example, GOF mutations are more common than LOF alleles for patients with paroxysmal non-kinesigenic dyskinesia (PNKD), while non-PKND movement disorders were observed in LOF mutations [230]. In a GOF knock-in murine model carrying a human BK-D434G mutation, mice showed increased susceptibility to generalized seizures and motor deficits related to hyperexcitability in cortical pyramidal and cerebellar Purkinje neurons. Importantly, blocking BK channels with paxilline suppressed seizures, motor disturbances, and neuronal hyperexcitability [155]. Similarly, a D434G mutation that produces an increase in Ca^2+^ sensitivity that potentiates BK channel activity [231] was described in individuals affected by epileptic seizures and/or PNKD [232]. Moreover, a rare disease called KCNMA1-linked channelopathy, is also characterized by seizures and abnormal movements associated with the clinical detection of a KCNMA1 variant [232]. Another mutation, the N995S variant, is present in individuals with epilepsy but not PNKD. Such a mutation in the RCK2 domain increases BK channel currents by increasing voltage sensitivity but does not affect Ca^2+^-dependent activation. In BK-N995S/β4 channels, the effect of the mutation is slightly more pronounced [233]. However, in a mouse model with the N995S mutation, PNKD-like symptoms could appear after stress exposure [232].

We recall here that targeted deletion of the *KCNMB4* gene coding for β4 subunits narrows action potentials and increases firing in dentate granule cells of the hippocampus, possibly leading to an enhancement of transmitter release and temporal lobe seizures [72]. Consistent with the involvement of the β4 subunit in epilepsy, pilocarpine-induced seizures generated a decrease in β4 subunits in dentate gyrus granular cells and a reduction in type II BK channels [234]. While β4 subunits seem to play an important role in epilepsy, a mutation in the gene encoding the accessory β3 subunits (KCNMB3b β3b-truncation) have also been frequently observed in patients affected by idiopathic generalized epilepsy [235].

Functional alterations of BK channel expression are also linked to neuropathic pain. A downregulation of BK channels in dorsal root ganglions and the spinal dorsal horn is accompanied by neuropathic pain [236,237,238]. Nerve injury promotes an increase in N-type Ca^2+^ channels in the spinal dorsal horn mediated by an upregulation of Cav2δ subunits, increasing pain signaling. Expressing the BK channel N-terminus, a domain that binds the Cav2δ subunit, promotes a reduction in the expression of N-type Ca^2+^ channels and analgesia [239]. Additionally, the pharmacological blockade of spinal BK channels with iberiotoxin produces mechanical hyperalgesia. In contrast, activation of the BK channel reduces tactile allodynia and mechanical and thermal hyperalgesia, in nerve-ligated rats [236]. Similarly, decreasing spinal β3 subunit expression reportedly attenuated neuropathic pain produced by nerve injury in mice [240], further supporting a role for BK channels in regulating pain.

BK channels are also involved in motor and cognitive diseases. For instance, in homozygous mice lacking α subunits, muscular tremors, muscular weakness, and gait disturbances have been reported. However, in an open field test, these animal models showed normal distance for locomotor behavior. In addition, these mice also display deficits in learning, with a delay in acquiring a new task but not in memory once the task was learned [241]. Similarly, a mutation (BKG354S) in the α subunit has recently been reported in a child with congenital and progressive cerebellar ataxia with cognitive impairment. This mutation dramatically reduced single-channel conductance and ion selectivity and reduced neurite outgrowth, cell viability, and mitochondrial content [242].

Besides humans, GOF mutations are also present in *Drosophila.* There, the SLO E366G mutation (equivalent to the human mutation D434G) generates motor disturbances in both larvae and adult stages [243]. In *C.elegans*, deleting the Shank gene Shn-1, a synaptic scaffolding protein related to autism, produces a decrease in BK channel currents and clustering at muscles and neurons [244]. Interestingly, BK channels may play a role in the physiopathology of fragile X syndrome [245] as accessory β4 subunits interact with FMRP proteins, whose loss causes the fragile X syndrome, characterized by intellectual disabilities associated with language deficits, hyperactivity, autistic behavior, and seizures. In the absence of β4 subunits, FMRP decrease channel activation and deactivation rates, but in the presence of β4, FMRP increase channel opening [81]. BK channel agonists have a major impact on rescuing social and cognitive impairments [246]. However, the diversity of BK channel expression within different neuronal populations and brain circuits and the poor selectivity of drugs, suggest that one should take the use of these compounds in clinical practice with caution.

A role for BK channels in modulating learning and memory processes also has been suggested [160,247,248,249]. For example, if a methyl-donor-rich diet is supplied to mice before mating, the offspring show disturbances in hippocampus-related memory tasks and long-term potentiation of synaptic transmission related to decreased β2 subunit expression in the hippocampus [250]. Additionally, anxiety and cognitive disturbances induced by thalidomide were correlated to increased BK channel activity. These alterations can be rescued by the BK channel blocker paxilline [251]. Similarly, BK channel currents are reduced in pyramidal neurons of the anterior cingulate cortex in a mechanical allodynia model generating anxiety behaviors in mice. Upregulated expression of β4 subunits in presynaptic and postsynaptic domains and increased neuronal excitability and presynaptic neurotransmitter release accompanied this phenomenon. BK channel activation with NS1619 reversed anxiety-like behaviors and the mechanical pain threshold [73]. Last but not least, a genome-wide association study reported that a single-nucleotide polymorphism (SNP) in the gene encoding α subunits (*KCNMA1*, rs16934131) was associated with late-onset Alzheimer’s disease (AD) risk in humans [252].

On the other hand, an SNP in the gene encoding β2 (*KCNMB2,* rs9637454) was strongly associated with hippocampal sclerosis, a comorbid neuropathological feature of AD [253]. While the functional consequence of these point mutations is unknown, rodent AD models have shed some light on how BK channels may contribute to AD pathogenesis. For example, in TgCRND8 mice; a mouse model for AD which overexpresses the amyloid precursor protein due to containing a double human amyloid precursor protein mutation; a decreased basal transmission associated with a reduced decay of afferent volleys has been reported. This difference was recovered when blockers for BK channels were present [254]. Moreover, intracellular injection of amyloid-β1–42 (Aβ1–42) peptides into neocortical pyramidal cells suppresses BK channel activity, which broadens evoked AP spike width and indirectly enhances Ca^2+^ influx [255]. Similarly, Ab42 oligomers, stimulate spontaneous transient Ca^2+^ release from RyRs. RyRs activate presynaptic BK channels and reduce AMPA-mediated neurotransmission in hippocampal neurons [256]. Conversely, in the triple transgenic AD mouse model 3xTg, increased basal transmission and pyramidal excitability and decreased synaptic plasticity at the hippocampal CA1 area were related to a hypofunction of BK channels. 3xTg also displayed poor performance in non-spatial and spatial memory tasks, a dysfunction that was improved when a BK channel activator was delivered through ventricular injection [257]. While this evidence suggests a potential role of BK channels in the pathophysiology of AD, a direct link between Aβ42 oligomers and BK channel activation at the early stages of AD remains to be identified.

## 8. Conclusions and Future Directions

Increasing evidence indicates that, in addition to its well-known role in regulating neuronal excitability, BK channels also regulate dendritic excitability and synaptic transmission in different brain circuits. Their presynaptic and postsynaptic expression within a specific synapse suggests they are also well positioned to regulate synaptic function of excitatory and inhibitory neurons. This effect may vary from synapse to synapse (i.e., either increase or decrease) and might also regulate synaptic plasticity. While indirect evidence suggests that BK channels can regulate long-term potentiation and depression, whether BK channels play a pivotal role in the induction or expression of a different form of synaptic plasticity needs further investigation. Similarly, their expression in glial cells has yet to be entirely understood. For instance, the role of BK channels regulating gliotransmission, thereby modulating synaptic brain function, remains largely unexplored. Future work will expand our understanding of the role of glial BK channels, particularly their contribution to regulating GABA homeostasis and signaling, as they have been reported to directly interact with the GABA transporter in astrocytes [207]. Notably, lipid molecules, including AA derivates and endocannabinoids, are known to stimulate BK channels in heterologous and peripheral systems. However, whether these lipids regulate brain function and behavior in a BK-dependent manner remains unknown. Given the ubiquitous expression of BK channels throughout the brain, it is not surprising that they have been implicated in several neurological disorders, such as epilepsy, mental retardation, autism, AD, anxiety, movement disorders, and chronic pain [229]. Using LOF or GOF animal models has increased our current understanding of the role of BK channels in the etiology of these disturbances. However, further work is required for better development of novel therapeutic strategies and interventions that may ameliorate these devastating disorders.

## Figures and Tables

**Figure 1 ijms-24-03407-f001:**
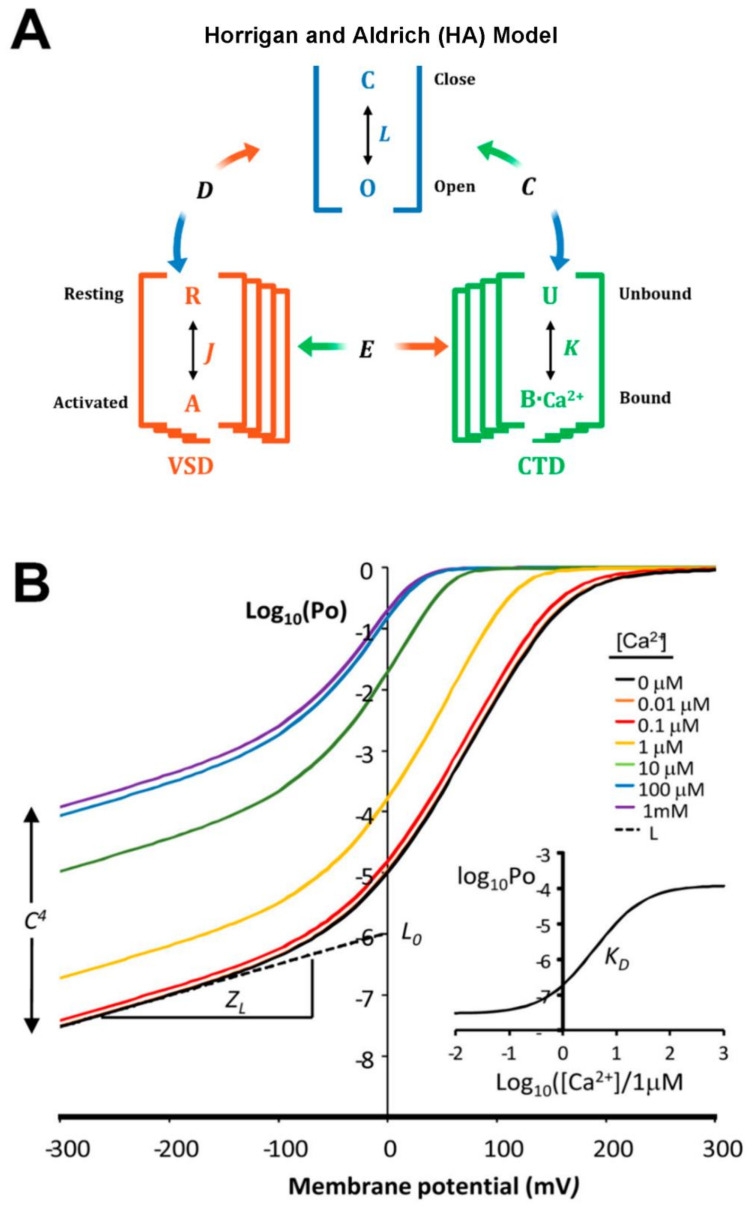
**BK Channel allosteric gating mechanism.** (**A**) H–A allosteric model [43]. Voltage sensor (R–A), Ca^2+^ binding, and C–O transitions are defined by the equilibrium constants *J*, *K*, and *L*, respectively. Voltage sensors and Ca^2+^ sensors are coupled to the pore by allosteric factors D and C, respectively, and coupling between sensors is performed by *E*. (**B**) Simulated data of the Log10 of the probability of opening vs. voltage obtained using the H–A model. Notice that when all the voltage sensors are at rest the parameters *L*, *zL*, and *C* can be obtained.

**Figure 2 ijms-24-03407-f002:**
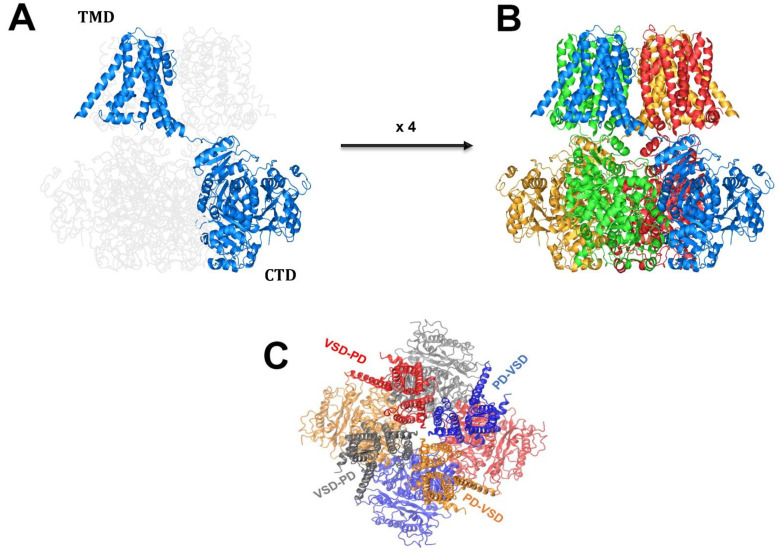
**Structural characteristics of the BK channel.** (**A**) The α subunit is formed by a transmembrane domain (TMD) composed of seven α helices (S0–S7) and a large carboxy terminal (CTD) containing two regulatory potassium conductance domains. (**B**) The BK channel is a tetramer formed by 4 α subunits in which the CTD acquires a swapped conformation and the 4 CTDs form the gating ring. (**C**) Top view of the BK channel. The voltage sensor domain (VSD) and the pore domain (PD) have been colored to highlight the non-swapped configuration of the VSD and the PD.

**Table 1 ijms-24-03407-t001:** Functional expression of BK channels in the central nervous system.

CNS Area and Neuronal Type	Site of Neuronal Expression	Function	Subunit and/or Pharmacology	References
Prefrontal cortex pyramidal neurons	Pre- (AIS and axon) and postsynaptic (soma)	Limit AP broadening and allow high-frequency firing.	IBTX-sensitive	[151,152]
			Pax-sensitive	[153]
Visual cortex pyramidal neurons	Postsynaptic (basal dendrites)	Reduce EPSPs in small-headed spines.	α subunit, Chtx-sensitive	[25]
Somatosensorial layer 5 pyramidal neurons	Postsynaptic (soma and dendrites)	Reduce influx of ions through NMDAR and increase threshold for plasticity. Modulate dendritic Na^+^ and Ca^2+^ spikes.	IBTX and Pax-sensitive	[154]
			Pax-sensitive	[112,155,156]
Hippocampal CA1 pyramidal neuron	Pre- (axon) and Postsynaptic (soma and apical dendrites)	Limit repetitive firing of dendritic calcium spikes. AP repolarization and fAHP.	α subunit	[141,157]
			Chtx-sensitive	[158,159]
			Pax and IBTX-sensitive	[160]
			IBTX-sensitive	[137,142]
Hippocampal CA3 pyramidal neuron	Pre- (Axonal terminals) and postsynaptic (soma)	Decrease glutamate release. Regulate neuronal excitability.	α subunit	[161]
			β4 subunit	[15,162]
			Chtx-sensitive	[163]
			IBTX-sensitive	[23]
			IBTX and Pax-sensitive	[24]
Hippocampal dentate gyrus granule cells	Pre- (mossy fiber boutons and perforant path terminals) and postsynaptic (soma and dendrites)	Control perforant path induce EPSP. Regulate excitability. At mossy fiber boutons contribute to AP repolarization when Kv channels are blocked.	α subunit	[161]
			β2 and β4 subunits	[164]
			β4 subunit	[72]
			Pax-, Chtx- and IBTX-sensitive	[99]
			Pax-sensitive	[27]
Piriform cortex pyramidal neuron	Postsynaptic (soma)	Reduce excitability modulating odor perception.	Pax-sensitive	[165]
Anterior cingulate cortex pyramidal neurons	Pre- (axonal terminals) and postsynaptic (soma)	Reduce hyperexcitability induced by neuropathic pain state.	β4 subunit.	[73]
Somatosensorial SOM+ interneurons	Pre- (axon) and postsynaptic (soma)	Contribute to AP repolarization.	IBTX-sensitive	[166]
Amygdala central nucleus gabaergic neurons	Presynaptic	Limit GABA release.	β2 and β4 subunits	[12]
Substantia nigra pars reticulata gabaergic neurons	Postsynaptic (soma)	Contribute to hyperpolarization.	Chtx- and Pax-sensitive	[167]
Substantia nigra dopaminergic neurons	Postsynaptic (soma)	Not tested.	IBTX-senstive	[134]
Infragranular pyramidal neurons	Postsynaptic (soma and apical dendrites)	Not tested.	Chtx-sensitive	[135]
Lateral habenula nucleus neurons	Postsynaptic (soma)	fAHP increased by corticotropin-releasing factor activation.	IBTX-sensitive	[168]
Medial habenula nucleus neurons	Not described	Not tested.	LRRC55 γ subunit	[169]
Cerebellum Purkinje neurons	Presynaptic (paranodal junctions of axons) Postsynaptic (soma and dendrites)	Decrease dendritic Ca^2+^ spikes avoiding DSE propagation. Regulate afterdepolarization. In myelinated axons allow high-fidelity firing of AP.	α subunit	[161]
			β4 subunit	[170]
			Penitrem A sensitive.	[171]
			IBTX-sensitive	[172]
			IBTX–sensitive + IBTX-insensitive component	[138]
			IBTX and Pax-sensitive.	[102]
			Pax-sensitive	[173]
Cerebellar basket cells	Presynaptic (axonal terminals)	Not tested.	α subunit	[161]
Cerebellar granule cells	Postsynaptic (soma)	Contribute to hyperpolarization and AHP.	IBTX-sensitive	[170]
Suprachiasmatic nucleus neurons	Postsynaptic (soma)	Increase neuronal firing at night but decrease firing during day.	β4 subunit	[174]
			β2 subunit	[86]
Supraoptic nucleus neurons	Pre- and postsynaptic (dendrites, soma and nerve terminals)	β4 but not β1 allows potentiation by ethanol.	β1 subunit in soma and dendrites; β4 in nerve terminals	[79]
Medial vestibular nucleus neurons	Postsynaptic (soma)	Contribute to excitability. Can be activated by CaMKII and inhibited by PKC.	Not tested	[125]
Dorsal cochlear nucleus cartwheel cells	Postsynaptic (soma and dendrites)	Control EPSP.	IBTX-sensitive	[113,175]
Retinal ganglion cells	Postsynaptic (soma)	Regulate cell excitability.	IBTX-sensitive	[176]
A17 amacrine cells	Postsynaptic (varicosities)	Limit Cav channel activation and GABA release.	β2 subunit, IBTX-sensitive	[22]

Abbreviations: AIS: axonal initial segment; IBTX: iberiotoxin; Pax: paxilline; Chtx: charybdotoxin; EPSP: excitatory postsynaptic potential; AP: action potential; AHP: afterhyperpolarization; DSE: depolarization-induced suppression of excitation.

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
