# Peer review of "Ca2+- and Voltage-Activated K+ (BK) Channels in the Nervous System: One Gene, a Myriad of Physiological Functions"

_ijms, 2023, doi:10.3390/ijms24043407_

Round 1
Reviewer 1 Report
This manuscript by Ancatén-González et al is excellent. It comprehensively reviews all current knowledge about BK function in the nervous system, providing the field with a valuable updated reference. Although many questions still remain open, as clearly and honestly stated along the manuscript, our knowledge about the role of BK in the nervous system is continuously growing, and significant advances have been found in recent years, so this review is timely and there is no doubt that it will be very well received in the field. I have only a few very minor comments that I provide with the honest hope that they help to improve the manuscript. They are mainly typographic errors and a comment about editing in Table 1. I would also suggest to revise some citations as suggested to make it easier for readers to find them, if needed.
Line 90: “Code” should be “coded”
Line 104: Revise. “one kinetic scheme the BK”…maybe add “for”?
Line 135: It could perhaps be reminded here that the channel has large conductance, which is also crucial for it to be a “damping machine”
Line 244: “in neurons fmpr knockout” should probably be “neurons from …”
Line 317: Reference is cited as “J. Zhang et al. 2018”. Please revise
Line 522: “rods” should be “rod”
Line 615: Reference is cited as “Mi Park et al, 2022”. Please revise (note that in line 620 it is correctly cited as “Park et al 2022”, revise if this has lead to incorrect duplication of references)
Line 768: “understand” should be replaced by “understanding of”
Line 770: “ameliorated” should be “ameliorate”
Line 1020: The cited reference seems incomplete, it does not include the full reference (journal name, etc): Gómez, R., Maglio, L. E., & Gonzalez-hernandez, A. J. (2021). NMDA receptor – BK channel coupling regulates synaptic plasticity in the barrel cortex. It would be adviced to replace it by the full reference that can be found at https://pubmed.ncbi.nlm.nih.gov/34453004/
A final comment: Table 1 is very useful and provides a significant update that will be very well received in the field. However, column 4 (subunit and/or pharmacology) seems slightly confusing. Are the items related to the papers in column 5, or is this just a list of the subunits and pharmacological features that have been globally reported in reference with the referred areas? In any case, a simple edit adding bullets to the list will greatly help to make it more readable. This is a very minor thing, of course.
Reviewer 2 Report
The authors present an up-to-date review of the importance of neuronal BK channels. Very nicely presented is the description of the biophysical functioning and the processes of structural changes required for the opening and closing of the ion channel in front of its discrete voltage and calcium regulation. This part of the review article is reinforced in its comprehensibility by two appropriate figures.
The subchapters "BK Auxiliary Subunits”, “BK Channel Coupling with Ca2+-Permeable Channels” and BK channels dysfunction in neurological disorders" are also impeccably presented and up-to-date.
However, regarding the central subchapter "BK channels differentially contribute to control brain function", one would have wished for more sorting, e.g. according to a brain anatomical classification following Tab. 1. Also, opposing effects of the BK channel on neuronal activity should be further elaborated here.
In my opinion, the review article should become clearer and more understandable on this latter point before acceptance.
Minor notes:
Line 203: Reinhart vs. Reinhard
Line 244: "of" is missing
Line 674-676: sentence requires rephrasing
